# A Nationwide Digital Multidisciplinary Intervention Aimed at Promoting Pneumococcal Vaccination in Immunocompromised Patients

**DOI:** 10.3390/vaccines11081355

**Published:** 2023-08-11

**Authors:** Shirley Shapiro Ben David, Orna Shamai-Lubovitz, Vered Mourad, Iris Goren, Erica Cohen Iunger, Tamar Alcalay, Angela Irony, Shira Greenfeld, Limor Adler, Amos Cahan

**Affiliations:** 1Health Division, Maccabi Healthcare Services, Tel Aviv 6812509, Israelirony_an@mac.org.il (A.I.); adler_l@mac.org.il (L.A.); 2Tel Aviv School of Medicine, Tel Aviv University, Tel Aviv 6139001, Israel; 3Infectious Diseases Unit, Samson Assuta Ashdod University Hospital, Ashdod 774762, Israel; amosc@assuta.co.il

**Keywords:** pneumococcal vaccine, intervention, digital, immunocompromised, electronic medical record, PCV13, PPSV23, alerts

## Abstract

Immunocompromised patients (IPs) are at high risk for infections, some of which are vaccine-preventable. The Israeli Ministry of Health recommends pneumococcal conjugate vaccine 13 (PCV13) and pneumococcal polysaccharide vaccine 23 (PPSV23) for IP, but vaccine coverage is suboptimal. We assessed the project’s effectiveness in improving the pneumococcal vaccination rate among IP. An automated population-based registry of IP was developed and validated at Maccabi Healthcare Services, an Israeli health maintenance organization serving over 2.6 million members. Included were transplant recipients, patients with asplenia, HIV or advanced kidney disease; or those receiving immunosuppressive therapy. A personalized electronic medical record alert was activated reminding clinicians to consider vaccination during IP encounters. Later, IP were invited to get vaccinated via their electronic patient health record. Pre- and post-intervention vaccination rates were compared. Between October 2019 and October 2021, overall PCV13 vaccination rates among 32,637 IP went up from 11.9% (*n* = 3882) to 52% (n = 16,955) (*p* < 0.0001). The PPSV23 vaccination rate went up from 39.4% (12,857) to 57.1% (18,652) (*p* < 0.0001). In conclusion, implementation of targeted automated patient- and clinician-facing alerts, a remarkable increase in pneumococcal vaccine uptake was observed among IP. The outlined approach may be applied to increase vaccination uptake in large health organizations.

## 1. Introduction

Pneumococcal disease is a serious infectious disease caused by the bacteria Streptococcus pneumoniae. It is a significant global health concern, leading to various clinical presentations, including non-bacteremic pneumococcal pneumonia and invasive pneumococcal disease, a severe and potentially life-threatening pneumococcal disease. Invasive pneumococcal disease occurs when the bacteria invade generally sterile sites in the body, causing infections such as bacteremia, sepsis, meningitis, and osteomyelitis. The annual incidence of invasive pneumococcal disease in the Israeli population during 2019 was 2.05, 8.42, and 24.03 per 100,000 population among patients aged 18–49 years, 50–64 years, and 65 years and above, respectively [1]. Serotypes 12F, 8, and 3 were the predominant in causing invasive pneumococcal disease in Israel from July 2016 through June 2019 [1]. Immune compromise, caused by immunosuppressive treatments or underlying conditions, predisposes patients to pneumococcal disease due to diminished immune system function. In these patients, the burden of invasive pneumococcal disease, including disease incidence, hospitalizations, intensive care unit admissions, and mortality, is higher than in other patients [2,3]. A study from Canada reported a 12-fold higher incidence of invasive pneumococcal disease in immunocompromised compared to immunocompetent individuals, as well as a higher fatality rate [4]. Indeed, higher susceptibility to was found among transplant recipients [5,6], patients living with HIV [7], patients with hematologic malignancies [8], patients who use immunosuppressive medications [9], and asplenic patients [10], as well as patients with advanced kidney disease [9].

Vaccination is the most effective intervention for reducing the burden of pneumococcal disease. A large proportion of invasive pneumococcal diseases are vaccine-preventable [11,12]. Given the high susceptibility of immunocompromised individuals to pneumococcal disease, it is imperative to prioritize vaccination in this population. Studies have shown that a reduction in the incidence of invasive pneumococcal disease in adults, including immunocompromised individuals, after the introduction of the pneumococcal vaccines, pneumococcal conjugate vaccine 13 (PCV13) and pneumococcal polysaccharide vaccine 23 (PPSV23) [3,13]. In order to mitigate the impact of pneumococcal disease and improve overall public health outcomes, healthcare providers and public health authorities need to continue advocating for and implementing vaccination.

Israeli guidelines recommend vaccinating patients with immunosuppression against pneumococcus with both vaccines, PCV13 and PPSV23 [14]. In Israel, vaccines are almost exclusively administrated in the outpatient setting, through one of four Health Maintenance Organizations (HMO), usually at local nurses’ or physicians’ clinics nationwide. All vaccination are documented in the HMO centralized electronic medical record (EMR). Despite national recommendations, previous studies have shown a suboptimal pneumococcal vaccination rate among immunocompromised individuals, as only 11.3% of over 32,000 eligible patients had received PCV13, and only 39.4% had received PPSV23 [15].

The low vaccination rate can be attributed to various factors. One significant factor is the lack of awareness among healthcare providers regarding patients’ immunocompromised status and the recommended vaccines for this specific population [16,17]. This lack of knowledge may lead to missed opportunities for vaccination. Patients themselves may also be unaware of the availability and importance of vaccination. They might not realize that they are at higher risk due to their immunocompromised state, and thus may not actively seek vaccination [18]. Healthcare providers also play a crucial role in advising and encouraging patients to get vaccinated. Lack of physician recommendation was found as a strong predictor of low vaccination rate [19,20,21,22]. Addressing these factors and increasing awareness among healthcare providers and patients about the need for vaccination in immunocompromised individuals is essential to improve vaccination rates and protect this vulnerable population from preventable infections.

To address care gaps, in October 2019, a comprehensive intervention to elevate pneumococcal vaccination uptake among adult over 18 years was commenced in Maccabi Healthcare Services (MHS), the second-largest HMO in Israel. The intervention targeted clinicians as well as patients. This study aimed to assess the change in pneumococcal vaccination rate among immunocompromised individuals following the intervention. Here, we report the intervention and its outcomes.

## 2. Materials and Methods

### 2.1. Setting

MHS is one of four nonprofit HMOs in Israel. MHS serves over 2.5 million patients, representing about a quarter of the Israeli population. Of the 5500 physicians employed nationwide, about 1600 are primary care physicians. MHS has had an extensive centralized EMR system since the mid-1990s. National unique identification numbers assure data continuity.

Embedded within the MHS’ medical record is a provider-facing alert-based computerized decision support system. Implementing the five Rights of a computerized decision support system [23] provides clinicians with real-time clinical recommendations: *1. The right information:* recommendations are based on professional international and/or national clinical practice guidelines. *2. The right person*: alerts are presented to physicians responsible for assuring their patients receive optimal preventive medical care, and are licensed to act according to the system recommendations. *3. The proper format*: recommendations applicable to a patient can be seen by clicking a button in the patient’s EMR. *4. The right channel:* in addition to presenting recommendations in the EMR, most of the recommendations are also presented on MHS members’ health record, empowering them to take responsibility for their health. *5. The right time:* as noted, recommendations are triggered to pop up when an encounter is initiated in the EMR. If the physician cannot attend to the recommendations during the encounter, she can opt to show them again during the subsequent encounter. The MHS computerized decision support system data infrastructure allows new rules for alerts to be created independent of software version updates. Since 2009, more than 100 rules in various fields have been developed and added to the system, with high clinical utility [24]. These rules mainly address screening tests as part of preventive medicine, but also apply for monitoring of chronic conditions.

As previously reported, an automated immunocompromised patient (IP) registry has been developed based on this data [15]. Briefly, five categories of adults over 18 years of immunocompromised populations were included in the registry: patients receiving immunosuppressive therapy; patients living with HIV; solid organ and bone marrow transplant recipients; patients with advanced chronic kidney disease; and patients with asplenia.

Recommended in those patients are one dose of PCV13 as well as one dose of PPSV23, plus a booster dose after five years before the age of 65 and/or one dose of PPSV23 after the age of 65. At the time before the intervention, PPSV23 was available without a prescription or referral from a physician, free of charge. PCV13 required pre-approval and prescription, and was available for purchase at a subsidized price. Patients with cancer, dialysis patients, and PLWH were entitled to receive the vaccine free of charge. Of note, PPSV23 was available from the year 2000, and PCV13 from 2016. PPSV23 was part of the national program for quality care in the community for the elderly (>65 years) [25], with EMR alerts for providers and active reminders for patients by text messages and phone calls.

### 2.2. Intervention

Starting in October 2019, a nationwide quality improvement project was implemented to improve pneumococcal vaccination uptake among IP. The target immunocompromised population was identified by the IP registry [15]. The intervention included three steps. First, the need for preapproval for PCV13 is waived in eligible IP. Second, a real-time EMR pop-up alert computerized decision support system for reminding pneumococcal vaccine triggered during IP encounters was introduced. Third, an active patient-targeted health promotion campaign for pneumococcal vaccination was launched.

During an eligible patient visit (including telemedicine), physicians and nurses were prompted with an alert reminding them to consider providing a pneumococcal vaccine (Figure 1). The recommendation for the pneumococcal vaccine was based on the Israeli Ministry of Health vaccination guidelines in combination with previous vaccine uptake and the immunosuppressive disease, using ten different sets of codes for different scenarios (Figure 2). Access to health data from all encounters within MHS and some structured data (e.g., claims) from external providers for assures reliable system recommendations. Each recommendation was developed by a team of informaticians and clinicians in the MHS Medical Informatics department, who define rules for triggering vaccine alert pop-ups. Alerts underwent validation before implementation using a sample population. A physician provided direct feedback from the alert to the MHS Medical informatics team. Vaccine recommendations were accompanied by a short explanation so that users could understand the basis for any advice offered. Physicians had the option to accept a recommendation, decline or postpone it to the next visit (Figure 1). Acceptance would generate an automated referral for vaccination. Declining required documenting the reason (e.g., patient refusal or other medical considerations) and prevented the alert from being prompted on subsequent visits. Documenting vaccination also turned off the alert.

In May 2020, an active patient-targeted health promotion digital campaign was started. Eligible unvaccinated patients were actively invited to get vaccinated via their MHS digital patient health record (both desktop and mobile) and via personal text messages and e-mails (sent every six months).

### 2.3. Analysis

The primary outcome was vaccination with PCV13 and PPSV23. All IP included in the registry at the beginning of the intervention were followed until vaccination, discontinuing MHS membership, death, or end of the study period in September 2021. For overall rates, PCV13 and PPSV23 vaccination rates were compared during the pre- and post-intervention phases. Demographic characteristics were collected. Vaccination rates were calculated as a fraction of the actual vaccination number per eligible IP.

Demographic characteristics of the pre- and post-intervention IP groups were compared using the chi-square test, the Student t-test, or the Mann–Whitney test, depending upon the variable distribution. A comparison between pre- and post-intervention vaccination rates was performed using the chi-square test.

## 3. Results

During the period October 2019 to October 2021, overall PCV13 vaccination rates among IPs rose from 11.9% (n = 3882) to 52% (n = 16,955) (*p* < 0.0001) (Figure 3). The most prominent rise occurred among patients with advanced chronic kidney disease (6.8% to 51.9%) and asplenia (21.8% to 67.5%). PPSV23 vaccination rates went up from 39.4% (12,857) to 57.1% (18,652) (*p* < 0.0001). As of October 2021, 13,624 (41.7%) of IPs were up to date with their pneumococcal vaccination (both PCV13, and PPVS23) as opposed to only 2485 (7.6%) before the intervention.

Demographic characteristics of targeted PCV13 IPs are shown in Table 1. Patients who received PCV13 during the intervention were older, with a mean age (SD) of 58.9 (±16.9) versus 56.6 (±18) (*p* < 0.001), and were of higher socioeconomic status. These patients had more prolonged immunocompromised status, and had more comorbidities such as hypertension, diabetes, cardiovascular disease, and chronic obstructive pulmonary diseases. They also had more monthly primary care visits compared to those who did not get the vaccine (1.8 visits per month versus 1.1, *p* < 0.001). Minorities such as Arabs and Ultraorthodox Jews were under-represented among the vaccinated population. Demographic characteristics of targeted PPV23 are shown in Appendix A. Patients who received PPVS23 during the intervention were older, with a mean age (SD) of 54.9 (±14.7) versus 47.7 (±13.7) (*p* < 0.001). These patients also had more comorbidities and primary care visits per month.

Two peaks of PCV13 immunization were observed during the intervention. The first occurred during the last quarter of 2019 and coincided with the launch of EMR alerts (Figure 4a). During this period, the PCV13 vaccination rate rose from 11.9% to 23.1% (Figure 4b). After that, the pace of vaccination slowed, which coincided with the first COVID-19 pandemic lockdown. The patient-targeted digital campaign, which began in May 2020 (after the first COVID-19 lockdown was lifted), was followed by a second peak of vaccination. During December 2020, the vaccination rate dropped and stayed stable until the end of the study period (Figure 4a). Of note, this happened shortly after initiating the COVID-19 vaccination campaign. The PPSV23 immunization pattern remained relatively stable and showed a modest increase over time (Figure 4c). During the first year of the intervention, there was an average rise of 3% in immunization rates and an average rise of 1% during the second year of the intervention.

## 4. Discussion

This study describes a system-wide approach intervention to increase pneumococcal vaccination in a large health maintenance organization. This included clinician- and patient-targeted alerts and reminders implemented in a target population at high risk for invasive pneumococcal disease, identified using an automated IP registry. During the intervention, a substantial and statistically significant increase in pneumococcal vaccination was observed among IPs. PCV13 vaccination rates went up from 11.9% to 50.1% (*p* < 0.0001), and the PPSV23 vaccination rate rose from 39.4% to 57.1% (*p* < 0.0001). Of note, the provider-facing (EMR) as well as patient-facing (MHS digital patient health record) electronic alerts remain active since their launch, with the goal assuring vaccination of new eligible patients.

All vaccinations are meticulously documented in the HMO centralized EMR. Thus, the vaccination rates recorded in the EMR accurately reflect the actual vaccination rates. This comprehensive documentation system ensures that vaccination data is accurate, up-to-date, and readily accessible to healthcare providers and public health authorities. It also facilitates the monitoring and evaluating of vaccination programs, enabling better decision-making and targeted interventions to improve overall vaccination coverage among IP and other targeted populations.

EMR pop-up alerts reminders deliver actionable recommendations at the point of care [27]. They have been reported as an efficient tool to improve vaccination uptake [28]. Studies in small clinics demonstrated their value in improving pneumococcal vaccination among IPs [29,30,31]. A plausible explanation for this effect is that EMR alerts may help overloaded clinicians appreciate patient vaccination eligibility. As EMR alerts may disrupt encounter workflow and cause alert fatigue, the option of turning off the alert or receiving a reminder during a subsequent visit was added to the MHS intervention with minimal interference to clinicians’ workflows. Indicated by the meager rate of physicians requesting that the computerized decision support system be de-activated during their work with the EMR (unpublished data), our impression is that physicians generally welcome the computerized decision support system reminders, which help them to not forget to offer preventive health measures to their patients, and save time filling out referrals. As noted, implementing computerized decision support system recommendations was followed by increasing compliance with corresponding clinical guidelines [23].

We believe that the main factors contributing to compliance with the MHS computerized decision support system involve that clinicians in the development of alerts; reliance on current and complete, patient clinical information; allowing physicians to decide whether recommendations are presented in bulk for their patient population or during an encounter with a specific patient; and linking alerts with test ordering procedures to save clinician time.

Patients also play a crucial part in quality improvement interventions, including vaccination. As immunocompromised individuals may not be aware which vaccines are available to them, patient-facing reminders may help empower patients [21]. Other studies have found personal invitations and reminders directed at patients to be valuable tools to increase the vaccination rate [32,33].

The PCV13 uptake observed was greater than that of PPSV23, probably due to a higher PPSV23 baseline vaccination rate [15]. Of note, PPSV23 vaccination for the elderly is a national quality control measure in Israel, and EMR alerts for PPSV23 had been in operation for several years before the intervention period. As a result, the impact of the intervention on PPSV23 uptake was less pronounced compared to PCV13.

Older patients and those with comorbidities other than cancer were more likely to receive PCV13 during the intervention. A similar positive association between age and vaccination uptake was also found in other studies [34,35]. Comorbidities such as hypertension, cardiovascular diseases, diabetes, chronic obstructive pulmonary disease, and osteoporosis are more prevalent in the elderly, and contribute to the risk for invasive pneumococcal disease [36], which may contribute to higher physician commitment to, and patient compliance with, vaccination, as found in other studies [34,37,38]. Patients with cancer were less likely to get the vaccine. This may be due to concerns about side effects (especially with concomitant chemotherapy) or vaccine effectiveness during active anti-cancer treatment [39]. In line with other studies, lower socioeconomic status and minority groups were less likely to be vaccinated with PCV13 during the intervention [40,41]. Perhaps specific outreach strategies aimed at these populations or proactively scheduling an encounter for vaccination would have helped increase vaccination.

As of today, the scope of the IP vaccination intervention has been expanded to include meningococcal disease. The digital platform developed for the intervention was used later in the mass COVID-19 vaccination campaigns. With the introduction of PCV20 in Israel, a similar intervention will be relaunched with an updated set of rules for EMR alerts explicitly targeting the population of eligible patients, per the new guidelines issued by the Israeli Ministry of Health.

This study has several limitations that need to be acknowledged. Firstly, the reliance on an automated IP registry might have resulted in some immunocompromised patients needing to be identified and, therefore, not targeted as part of the vaccination campaign. Secondly, the study’s intervention design makes it challenging to establish a direct causal link between the intervention and the rise in vaccination rates. The observed increase in vaccination rates cannot be causally attributable to it. As the intervention was launched during the COVID-19 pandemic, patients’ fear might have increased adherence rates. Another limitation is the lack of data reflecting provider responses to the EMR alerts regarding whether patients accepted or declined the vaccination recommendations. This information could have provided valuable insights into the impact of the alerts on provider-patient interactions and decision-making regarding vaccinations. Further research and consideration of these limitations is essential for a comprehensive understanding of the program’s overall effectiveness and its potential implications for other similar interventions. In addition, further assessment is needed to evaluate clinical health outcomes, including the incidence of invasive pneumococcal disease during and after the study period in the vaccinated versus unvaccinated IP.

## 5. Conclusions

We report a substantial rise in pneumococcal vaccine uptake following the introduction of a comprehensive clinician- and patient-targeted digital intervention to promote vaccination in a population of IP. This is an effective method for increasing vaccination uptake in large health organizations. Additional measures to increase vaccination rates in vulnerable populations are needed.

## Figures and Tables

**Figure 1 vaccines-11-01355-f001:**
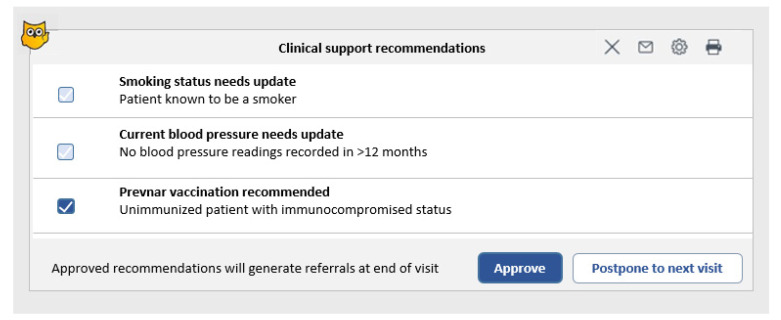
A targeted pop-up vaccine recommendation is displayed in the electronic medical record during a medical encounter with a specific patient. A screenshot of the MHS electronic health record clinical decision support system showing a targeted alerts pop-up screen activated during a medical encounter. Selecting the checkbox next to the recommendation automatically generates a referral for a nurse to administer the vaccine. The physician also has the option to have the recommendation presented again during the next visit.

**Figure 2 vaccines-11-01355-f002:**
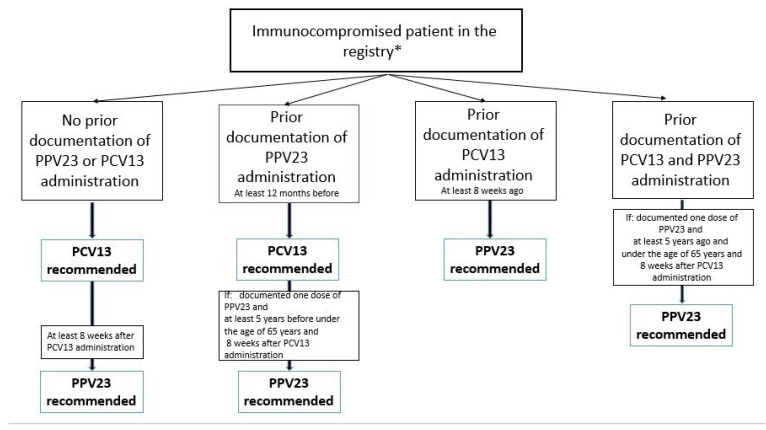
Algorithm for electronic medical record reminders for pneumococcal vaccines. The algorithm for pneumococcal vaccine electronic health record reminders involves using the EMR system to generate automated reminders for healthcare providers during patient encounters, specifically prompting them to assess and recommend pneumococcal vaccination based on patient criteria such as age, underlying medical conditions, and previous vaccination history * Excluded: bone marrow transplant, active oncologic patients. PCV13: pneumococcal conjugate vaccine; PPSV23: pneumococcal polysaccharide vaccine.

**Figure 3 vaccines-11-01355-f003:**
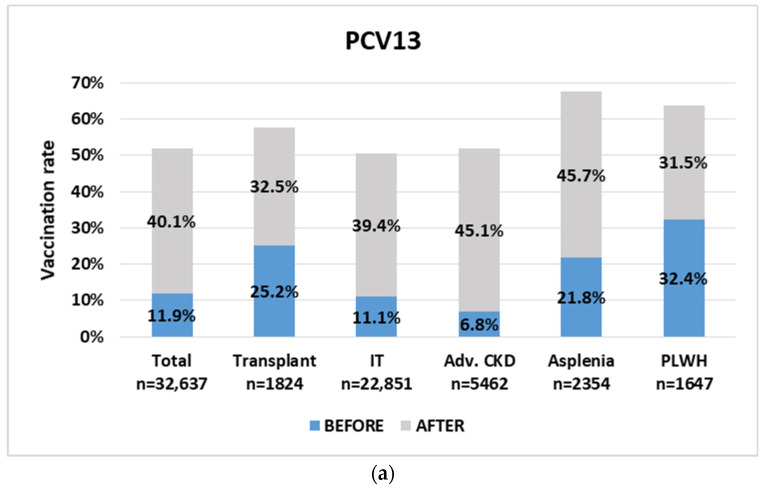
Vaccination rates among immunocompromised patients prior to and following the intervention. Vaccinated patients are included in the immunocompromised patients registry [15]. (**a**) PCV13: pneumococcal conjugate vaccine; (**b**) PPSV23: pneumococcal polysaccharide vaccine; IT, immunosuppressive therapy; Advanced CKD, chronic kidney disease (GFR < 30 mL/min); PLWH, patients living with HIV.

**Figure 4 vaccines-11-01355-f004:**
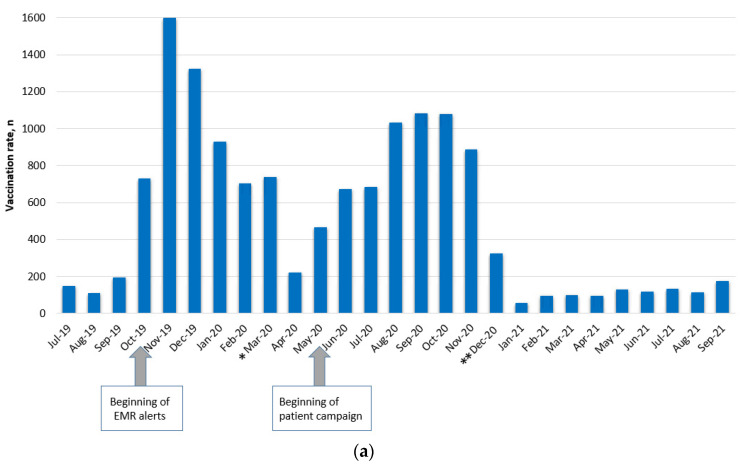
(**a**). Monthly PCV13 vaccinations during the intervention period. * Beginning of COVID-19 pandemic ** COVID-19 vaccination started. (**b**). The proportion of patients’ PCV13 vaccination over time. (**c**). Monthly PPSV23 vaccinations during the intervention period. * Beginning of COVID-19 pandemic ** COVID-19 vaccination started. (**d**). The proportion of patients’ PPSV23 vaccination over time.

**Table 1 vaccines-11-01355-t001:** Characteristics of the targeted intervention population for PCV13 administration.

Variable	Received PCV13 Vaccine during the Intervention (n = 13,073)	Not Receiving PCV13 Vaccine during the Intervention (n = 15,682)	*p* Value
Age (years)—mean (SD)	58.9 (16.9)	56.6 (18)	<0.001
Gender, Female—no. (%)	6571 (50.3)	8680 (55.4)	<0.001
Socioeconomic status ^‡^—no. (%)			
Low	2252 (17.2)	3366 (21.5)	
Med	6509 (49.8)	7599 (48.5)	
High	4312 (33)	4717 (30.1)	<0.001
Residential community			
General (Non Ultraorthodox Jews)	11,943 (91.4)	13,806 (88)	
Ultraorthodox Jews	591 (4.5)	931 (5.9)	
Arab	539 (4.1)	944 (6)	<0.001
Current Smoker—no. (%)	1659 (12.7)	2079 (13.3)	0.159
Comorbidities- no. (%)			
Hypertension	5974 (45.7)	5726 (36.5)	<0.001
Diabetes	3233 (24.7)	2977 (19)	<0.001
Cardiovascular disease	2185 (16.7)	1900 (12.1)	<0.001
COPD	7335 (5.9)	642 (4.1)	<0.001
Cognitive impairment	292 (2.2)	393 (2.5)	0.14
Cancer	1446 (11.1)	4442 (28.3)	<0.001
Osteoporosis	3066 (23.5)	2703 (17.2)	<0.001
IBD	1950 (14.9)	1595 (10.2)	<0.001
Time from entry to registry, (Yr)—mean (SD)	6.3 (5.6)	4.9 (5.1)	<0.001
Number of primary care visits per month, mean (SD)	1.8 (0.82)	1.1 (0.82)	<0.001
Family physician specialist/intern no. (%)	5741 (45.3)	6372 (42.9)	<0.001
Internal/General medicine specialist no. (%)	6946 (54.7)	8466 (57.1)	<0.001
Primary care physician age, mean (SD)	54.2 (9.8)	55.1 (10.4)	0.134

Yr, years; SD, standard deviation; CVD, cardiovascular disease; COPD, chronic obstructive pulmonary disease; IBD, inflammatory bowel disease. ^‡^ SES—socio-economic status; defined by the Israel Central Bureau of Statistics [26] as Low, 1–3; Medium, 4–6; High, 7–10.

## Data Availability

Not applicable.

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
