# Peer review of "A Nationwide Digital Multidisciplinary Intervention Aimed at Promoting Pneumococcal Vaccination in Immunocompromised Patients"

_vaccines, 2023, doi:10.3390/vaccines11081355_

Round 1

Reviewer 1 Report

The present study by David et al. depicts an assessment of the change in pneumococcal vaccination (PV) rate among immunocompromised patients (IP) following the intervention based on an electronic medical record alert, commenced in Maccabi Healthcare Services (MHS), Israel. Patients receiving immunosuppressive therapy; asplenic; people living with HIV; transplant recipients and patients with advanced kidney disease were included in the study. Following the implementation of targeted automated patient and clinicians alerts, a remarkable increase in pneumococcal vaccine uptake was observed among IP.

The authors have demonstrate the importance of such electronic data record based interventions to increase the vaccination rate in the vulnerable population. However, the study has following concerns, which needs to be addressed

1.       The introduction section lacks the epidemiological data of pneumococcal disease particularly in the targeted region and population.

2.       The data regarding susceptibility of IP towards the disease should be discussed to demonstrate the importance of vaccination in such population.

3.       Authors should also discuss on the possibility of fear factor, which led to increase in vaccination uptake in vulnerable population during COVID pandemic.

4.       Please include demographic data of PPSV23 in Table 1.

5.       There is a typo error in Mean age data of population received PCV13 vaccine during the intervention. Please correct.

6.       There is also inconsistency in labeling of pneumococcal polysaccharide vaccine in the manuscript and needs to be harmonised. 

7.       The manuscripts has numerous grammatical errors and need improvement.

English language needs to be improved upon as there are numerous grammatical errors.

Reviewer 2 Report

Dear authors of the manuscript "A nationwide digital multidisciplinary intervention aimed at promoting pneumococcal vaccination in immunocompromised patients".

This is an interesting subject about the most important issue in vaccinology: to really get the vaccine in those who needs it the most. The manuscript describes the Israeli experience of targeted promotional campaign of pneumococcal vaccination. The study is funded by Pfizer.

The manuscript is well written and comprehensive. I have a few comments/suggestions:

1. Please explain in the introduction how well is the vaccines given covered in the EMR and comment in the discussion: how well do the registered vaccinations (before the campaign) correspond to actual vaccines given? Is this known?

2. Please clarify if the campaign was adressed to both children and adults? In table 1 you can see that pediatric PCV13 were given to children but was this group targeted in the campaign?

3. Please provide for PPV23 the same graph (Fig 4a) as for PCV13 as I understand the campaign was for both vaccines?

4. Please comment something about the external validity outside Israel? Other similar experiences in other countries?

5. Please comment if changed recommendations in Israel now after introduction of PCV20? Will this campaign be relaunched?

6. Please give some information on incidence of IPD in Israel/in the targeted group and if there has been any change during the study period (although difficult since most countries had a steep decline in incidence during the pandemic years).

Minor comment:

Please use either PPV23 or PPSV23, in fig 3 it is written PPSV23?

Round 2

Reviewer 1 Report

Manuscript can be accepted.